# Digital Atlas of Tactics to Designing Sustainable Factories

Lia Marchi * and Ernesto Antonini

Department of Architecture, University of Bologna, 40136 Bologna, Italy; ernesto.antonini@unibo.it
* Correspondence: lia.marchi3@unibo.it

**Abstract:** For a long time, the design of factories has been profit-driven only, while their detrimental effects on the environment, perceptual-aesthetic interferences with the surroundings, and social disturbances on local communities have been largely neglected. Despite a growing attention towards these topics, literature shows that there is a fundamental knowledge and tool gap on design practices for holistically sustainable factories, and companies are often unaware of both negative and positive effects related to the impact of their sites on the landscape. This paper presents a toolkit that has been developed to support entrepreneurs and designers in devising more sustainable factories through an integrated perspective, which is the great novelty of the approach. The article focuses on one of its tools: a digital atlas of design tactics. These have been mapped in sustainable factories around the world and labelled with an ad hoc faceted classification. Each tactic is then described in an info-sheet, which feeds a web portal. There, the user is assisted in searching for the most suitable tactics and mutual links with other useful strategies. The main potentiality of the atlas is to encourage a holistic design approach by highlighting positive synergies among tactics from different fields.

**Keywords:** corporate image; green factory; good practices; faceted classification; landscape impact; LEED; New European Bauhaus



## 1. Introduction

Industrial sites are generally deemed as detractors of the living environment, even though the activities they host frequently contribute to sustaining the local economy.

Factories often affect their surrounding environmental quality by damaging the ecosystem's natural resources such as soil, water, and air. This includes, among others, harmful emissions because of energy and resource consuming processes, inefficient buildings, and transportation [1–3]. However, other aspects of the landscape are affected, but their importance is frequently overlooked: the perceptual-aesthetic and social qualities of the place. These refer, for example, to the obstruction of visual channels, fragmentation of the scenery, and disruption of the sense of place linked with the construction of a new factory [4–6].

The multi-faceted impact of factories on people's everyday landscapes has been largely ignored by designers and businesses. For decades, profit logics and function requirements have driven the design of industrial sites. Since the 1970s, interesting theories about green manufacturing and prototypes of green factories have progressively arisen [1,2,7–14]. Gradually, some interest for the aesthetic-perceptive and social disturbances of factories on the landscape has also grown, especially encouraged by a rising recognition of the favorable impact of factory architecture on corporate image [15,16], as well as of social-responsibility practices [17].

Despite this growing attention, the approaches that have been adopted until now are sectorial and focused solely on environmental sustainability. However, because mutual links exist between the various landscape dimensions, treating these aspects separately is no longer acceptable nor convenient. Instead, coupling good practices from the three spheres of the landscape may provide interesting benefits for both the firm and the territory.

Even the recently launched New European Bauhaus (NEB) stresses the potential and the need of approaching design in a more holistic way, by envisioning the future of the built

environment as 'beautiful, sustainable and together' [18]. Despite NEB not being focused on factories, it emphasizes, stronger than ever, that a good project is attractive, innovative, and human-centered, and that all the stakeholders ought to contribute to a better living environment. Consequently, it calls industries into action too.

A more holistic, sustainable approach to factory design emerges as the most appropriate way to support businesses in undertaking the necessary transition. This paper presents a study aimed at supporting companies and designers (e.g., architects, engineers, and urban planners) to design more sustainable factories, whether they are new constructions or renovations. In Section 1, already available means and approaches to this purpose and their weaknesses are discussed, and a toolkit developed by the authors to fill some of the detected gaps is introduced. In Section 2, the methodology used to develop one of the toolkit components is described, which is a Digital Atlas of Tactics to inspire the integrative design process in industrial sites. In Section 3, the results of the study are reported and later discussed in Section 4, along with limitations and further developments. Section 5 concludes the article by highlighting the potentialities of the Atlas.

### 1.1. Existing Design-Support Tools to Mitigate the Negative Impact of Factories on the Landscape

Lack of expertise and costs are recognized as the main barriers to the adoption of sustainable practices in industry [9,19]. On the one hand, companies often do not recognize the relevance of their impact and potential benefits of sustainability alike. Neither do they have an idea of what actions they can implement to mitigate the negative effects of their factories. On the other hand, small and medium enterprises, which constitutes 99% of European firms [20], struggle daily with resource shortages that make it difficult to address new investments, especially because sustainability is still perceived as an added cost rather than a value.

Thus, to mitigate the negative effects of industry on the environment, the scenery, and the local community, some tools have also been developed for factory design. On the one hand, environmental-impact assessment tools have spread to both identify room for improvements and communicate exemplary performances to the market. Thus, these tools were initially developed to fix the knowledge gap about environmental impact and their mitigation. Green Building Rating Systems (GBRSs) have had a remarkable success, because they aim at guiding the design team in improving the sustainability level of their project and, at the same time, they represent a marketing means for the business. By these, companies can also more clearly recognize economic benefits of sustainability (e.g., direct energy and cost savings, but also a gain of competitiveness due to an enhanced corporate image). Despite being initially conceived for housing and offices, today many international GBRSs offer schemes specifically targeted or general enough to be applied to industrial sites [21–25]. Nonetheless, these often focus solely on environmental impact [26]. Even when perceptual-aesthetic and social aspects are mentioned, they only refer to open spaces.

On the other hand, a few guidelines have been developed to support designers in making new industrial facilities sympathetic with their surroundings or mitigating the visual impact of existing factories. These are generally issued by local public authorities, or they are the outcome of international research projects [4,27,28]. Generally, these are focused on the sole visual aspect, targeted on local features, and lack an integrated approach with environmental and social effects.

Beyond these two main categories, other instruments are available to designers and entrepreneurs, such as conventional handbooks for factory design, which are gradually moving towards a more integrated design approach [29–32]. Likewise, there is also a number of architecture web portals where exemplary factory projects are included and serve as updated sources of inspiration [33–36].

Nevertheless, up to the authors' knowledge, there is no design-support tool that considers the multifaceted interferences of industries with the landscape, whilst policymakers and researchers stress that this combination could provide great benefits.

Furthermore, many of the available instruments are time-consuming, whereas businesses require quick, practical, and simple-to-grasp procedures. These should also inform clearly of the gains and potentialities of designing sustainable factories, in order to reduce common biases of the cost of the transition. To this end, promising design approaches could help, such as System Thinking and Integrative Design [37]: the first consists in considering the project and its parts as a series of relationships affecting each others; the second supports high-performance, cost-effective projects through an early and thorough analysis of the system and its interrelationships. In addition, digital technology is recognized as a powerful means to help designers and/or businesses in predicting and simulating the effect of their actions [12]. In fact, the same European Commission argues that digital technologies have a great potential for acting as enablers of the green transition, thus making the European Green Deal successful [38].

### 1.2. A Research Project Supporting a Holistic, Sustainable Design of Factories

Trying to fill the gap in tools for designing holistic sustainable factories, the authors have undertaken a research project with the aim of increasing the adoption of sustainable design practices in industrial sites. The main novelty of the study is addressing the relationship between factories and the landscape in all its complexity, hence considering together environmental, aesthetic-perceptual, and social interferences of manufacturing sites. To this end, a holistic design-support tool for sustainable factories has been developed, targeted directly on designers and businesses, representing the most original outcome of the research. The study was performed between 2016 and 2022, divided in two main research phases (Figure 1): the first is summarized and referenced in this paragraph; the second will be presented in this paper.

The study focuses on the agri-food industry, because more than other sectors it has both a heavy impact on and a strong relationship with the context. Food and beverage industries frequently host energy and resource-consuming processes that have detrimental effects on the environment. Moreover, due to historical requirements of proximity between farming and manufacturing, they are frequently located in rural or peri-urban settings where the perceptive and visual contrast with the scenery is sharper than elsewhere. At the same time, a tight, intangible relation between the environmental impact on the business and the image features of the factory in this sector, making the overall sustainability of the site critical. Briefly, the agri-food sector is where such criticalities can easily become opportunities for both the firm and the territory.

On this basis, given that a proper knowledge of the overall impact is required prior to perform any action, the study developed an impact-assessment system. The tool integrated the renowned U.S. Green Building Council LEED rating with a new category addressing perceptual-aesthetic interferences of factories with their surroundings. Eight specific credits were proposed to be added to the rating to this purpose. These range from measuring the morphological harmonization of the building with the scenery, to evaluating the site attractiveness of locals and visitors. Further information on the assessment system can be found in [39].

At the same time, a catalogue of good practices was developed to provide companies with inspiring examples to mitigate their impacts. These were retrieved from a collection of exemplary factories which have implemented good practices from at least one of the considered landscape dimensions (i.e., environmental, social, and perceptual-aesthetic). Practices—which are case-specific—have been organized via the LEED credit categories and then turned into general tactics. These latter are instead nearly universally valid. Further information on the catalogue can be found in [40]. Afterward, the iterative application procedure of the toolkit was defined, which is the combination of the assessment system and the catalogue of good practices, and it was evaluated in a case study.

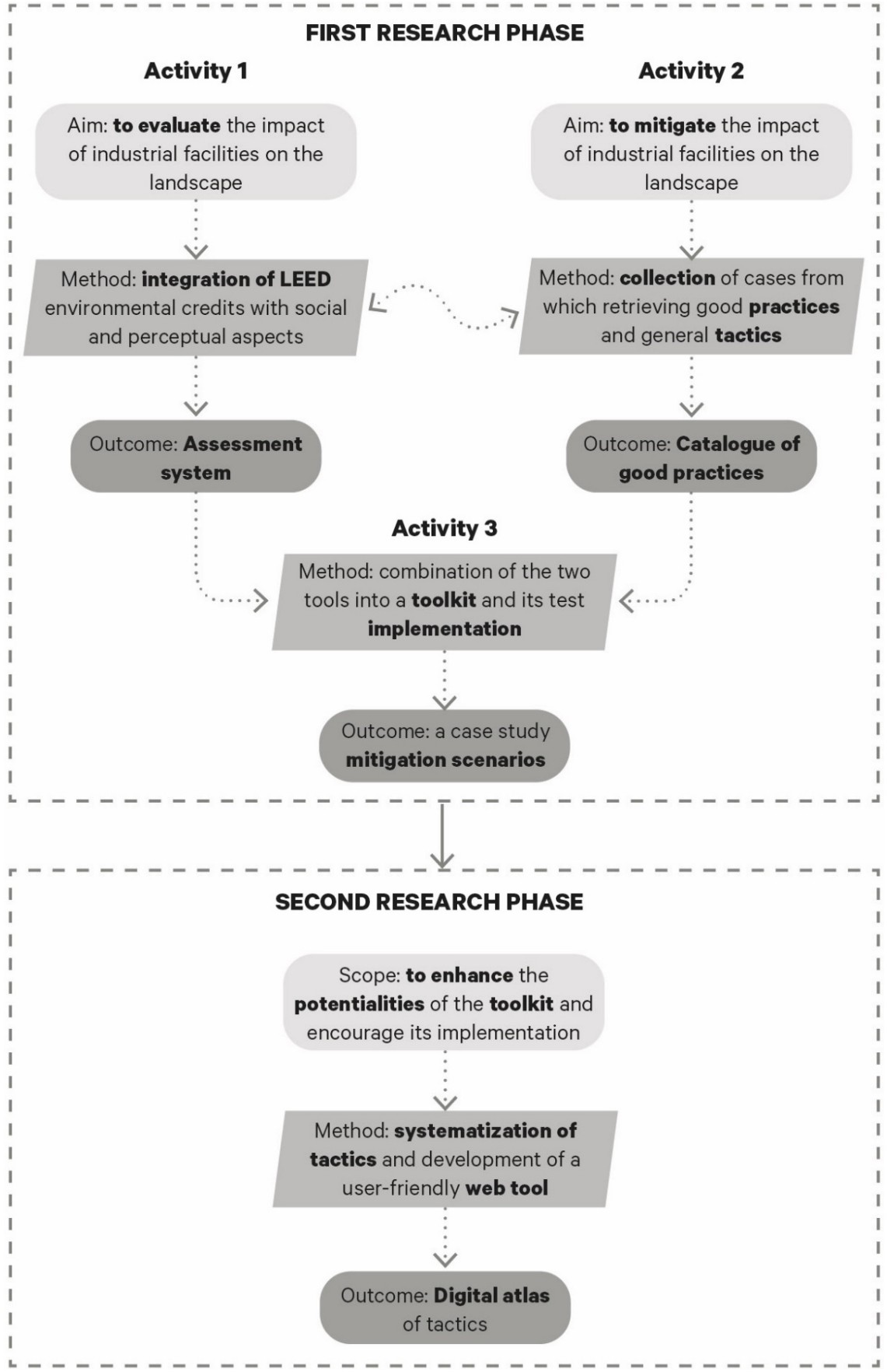

**Figure 1.** Overall research workflow.

The second phase of the research (2020–2022) aims at enhancing the potentialities of the toolkit, and especially at making it easy for businesses to access this knowledge database and thus implement mitigation tactics in their projects. Therefore, the goal of this research stage was to systematize the tactics to facilitate the user's search and implementation. Accordingly, the specific goals were:

- To make it possible for potential users to find and understand tactics suitable to their project peculiarities.
- To make the effectiveness of the tactic(s) in which they are interested clear to users.
- To highlight mutual synergies among tactics and thus promote holistic, sustainable design and related benefits.
- To make it simple to the final users to search for tactics through their own mental scheme (i.e., open, flexible, and multi-facet searches).

On this basis, the idea was to build a web portal, namely a Digital Atlas of Tactics, which could be flexible and dynamic enough to meet these requirements. Nowadays, a digital instrument has the potential to enable sustainability gains by managing complexity more effectively than a paper-based tool through, for example, algorithms or even Artificial Intelligence. In addition, a digital tool fits with the open nature of the research outcomes, as it allows to add new case studies, practices, tactics, or even categories at any time.

Therefore, the main novelty of the Atlas on which the paper focuses lies in its two key features: (i) the integrated approach to sustainability which underpins the whole research project and represents unique experimentation for factory design so far; and (ii) the use of the digital environment's opportunities to maximize the implementation potential of the research outcomes.

## 2. Materials and Methods

The development of the Atlas consists of the retrieval of tactics and their systematization onto a web site. Accordingly, the methodology consists of four main phases:

1. Collection of tactics, from the identification of eligibility criteria and the analysis of real-world case studies to the description of tactics.
2. Definition of a weighting matrix for tactics, which can guide the users in selecting the most suitable to their goals.
3. Definition of a faceted classification for tactics and its implementation, to highlight mutual links among tactics.
4. Definition of the architecture of the Digital Atlas to meet the needs and preferences of potential users.

### 2.1. Collection of Tactics

The first step consists in the selection of case studies from which tactics are retrieved. This entails revising the already selected cases from previous stages of the research project, but also finding new projects to expand the catalogue. The cases are found using keywords related to the concept of environmental, social, and economic sustainability of industrial sites in both scientific and commercial data sources. Boolean operators are used: for instance, green AND factory; brand OR corporate architecture; or landscape AND factory. Projects completed in the last twenty years are eligible, whether they are new constructions or restorations of existing sites. Provided that they are related to the food or beverage industry, no geographical or size limits are applied.

Once a case is selected, it is analyzed through a scheme which was developed during the first stage, whose further details can be found in [40]. In brief, the number of LEED credits which the case potentially addresses is marked and, if at least two credits are adopted, the case passes to the following stage. Then, each case is described onto a card that includes its basic facts (e.g., location, date, and type of intervention), references, and the adopted good practices. The latest are linked with the related LEED credits and recorded in the same groups into an MS-Excel spreadsheet.

Then, tactics are extracted from practices by two alternatives:

- When a practice recurs the same in at least two cases, it is converted into a tactic.
- When two practices have a common denominator, a general tactic is extracted.

Tactics can also be drawn from the scientific literature. A search in papers and design guidelines has been performed to this end, using the same keywords of the case study but looking for strategies valid in general for industrial facilities, not necessarily implemented in agri-food buildings. In this case, it is added as a new tactic or associated with the corresponding one from the practice.

Each tactic is assigned a code according to the following rule: the abbreviated code of the LEED credit to which it refers plus the progressive number of the tactic and its title. For instance, 'LT3.3—Adaptive reuse' derives from 'LT3' (the third credit of Location and Transport category) and it is the third tactic linked to that credit.

As the last stage, each tactic is described in specific columns of the spreadsheet, including: the tactic's description (e.g., what it consists of, benefits, and practical examples if available) and references to the case studies that it originated from and/or to the literature sources. Links among cells of credits, tactics, practices, case studies, and references in this stage allows a systematization of data for easy processing.

*2.2. Weighing Matrix*

The second phase is to define a matrix of objectives toward which the design of a sustainable factory should strive. Given that sustainability is often a matter of achieving the best compromise between issues of different natures, the specific scope of this matrix is to guide users through the Atlas in selecting tactics that provide them the best response to their needs and businesses' core values.

To this end, several international policy frameworks that are steering a sustainable change in many sectors were referred to, such as the Agenda 2030 for Sustainable Development [41], the European Green Deal [38], and the New European Bauhaus [18]. Their consistency with the research project was checked and their goals and strategic missions were investigated to find out if they could provide a set of objectives to measure the effectiveness of tactics. They were all too general for the scope of this study.

Therefore, the principles and objectives of the most diffused GBRSs for factories were investigated. In particular, DGNB (D), BREEAM (UK), CASBEE (J), and LEED (USA) were considered and their websites were accessed [24,42–44].

LEED was used as the main reference because its objectives are consistent with the overall framework of the research project. In addition, it comes from a set of a few but comprehensive impact categories associated with key indicators to measure how a project addresses each category [45]. For instance, the Climate Change category is associated with (i) Greenhouse Gas (GHG) Emissions Reduction from Building Operations Energy Use; (ii) GHG Emissions Reduction from Transportation Energy Use; and (iii) GHG Emissions Reduction from Materials and Water Embodied Energy Use.

From a detailed investigation of LEED categories, it emerged that, even though environmental assessment prevails over the other landscape dimensions, perceptual-aesthetic and social impacts are also addressed. Using some topic keywords, such as 'well-being' for social aspects and 'open space' for aesthetic issues, it was found that 12 out of 27 indicators address the former beyond environmental effects, and 6 address the latter.

The list of impact categories was then used to weight tactics in a way that is similar to the LEED point-allocation system [45], where each credit is assigned a score based on the relative contribution it gives to the impact categories. On this basis, the same weighting matrix of LEED (shown in Table 1) is assumed and applied to the retrieved tactics.

**Table 1.** Weighting matrix for tactics to repeat for each category. Parentheses show achievable points for each aspect.

| Relative Efficacy | Benefit Duration | Controllability of Effect |
|---|---|---|
| Negative (−1) | <1 year (0) | None (0) |
| None (0) | 1–3 years (1) | Occupants (1) |
| Low (1) | 11–30 years (2) | Operation and Maintenance (2) |
| Med (2) | 11–30 years (3) | Owner (3) |
| High (3) | >30 years (4) | Passive (4) |

However, the Atlas's purpose is not to rank projects but to support the design and guide users in mitigating the impact of their factories. So, the matrix itself is intended as an easy guide, and less-strict gauges than those of the LEED process were implemented. For the same reason, it was chosen that the contribution of each tactic to the impact categories be displayed on a Kiviat Diagram (or radar) rather than as a total point.

*2.3. Faceted Classification*

The goal of this phase is to define a classification system that can help users seek tactics fitting their project and needs. Furthermore, because holistic sustainability has been shown to bring interesting opportunities, the classification is also intended to encourage the adoption of tactics that benefit each other, therefore generating positive synergies.

The specific objectives of the phase are: (i) to identify classification categories that may be of interest to potential users of the Atlas; and (ii) to enlist accordingly a set of attributes for each category that may be the likely research keys utilized by designers and firms accessing the Atlas.

Therefore, the first stage was to study the diverse types of classification systems to select the most suitable to the research. A general overview of classification systems was performed and the two main types were identified, namely taxonomy and faceted classification [46]. In the former, the object to classify is included in one class that acts as a container, and classes of information are fixed a priori. In the latter, the object to be classified is broken down into independent attributes (i.e., facets); hence, classes are automatically generated by the selection of facets by the user (bottom-up classification) [47]. Multidimensionality, scalability, and flexibility of the faceted classification [48] make it more appropriate to the scope of the study.

On these premises, systems adopted in the building sector were investigated. In particular, the faceted systems most used nowadays are: Uniclass (United Kingdom), UniFormat, MasterFormat e OmniClass (United States of America), and DBK (Denmark) [49,50].

At the same time, classification systems used in architecture atlases and portals on the web were examined, because the architectural portal is very similar to the idea of Atlas that the study was aspiring for. The most accessed worldwide in 2021 [51] were analyzed, among which were ArchDaily, Achilovers, Divisare, and Architecture&Design [33–36]. Additionally, well-known search engines for architecture were investigated, such as Archi-INFORM and The Plan [52,53]. Each was surveyed for:

(i)   The system of classification (whether taxonomy, faceted, or mixed).
(ii)  The type of interface (direct selection of category/attribute, Simple Query Interface (SQI), multiple filters, or a combination of them).
(iii) The list of categories and attributes (when available, divided per level).

The analysis demonstrated that most of them are based on a faceted classification often combined with free search (SQI); hence, the faceted classification was confirmed as the most appropriate system for the study.

Basing on the categories retrieved from general literature on classification, buildings' classification systems, and the architecture portal analysis, a set of suitable categories and attributes for the Atlas were defined. The main references were, respectively, the thirteen general categories defined by the Classification Research Group (CRG) [54] and the ten

building-related categories of Uniclass2 [55]. The facets (categories) relevant to tactics were selected and a list of possible attributes was retrieved from the architectural portals' investigation, but also inductively from the list of tactics. A glossary with explanations of facets and attributes has also been defined to help the user in browsing the Atlas.

The last stage consisted in the implementation of the faceted system to the list of tactics retrieved in the first phase.

### 2.4. Architecture of the Digital Atlas

As final phase, the digital structure of the Atlas was defined to prepare its potential development on the web. The phase aims at defining how contents are organized, namely, the architecture of the information.

Because the primary scope of this operation is to facilitate the user browsing and retrieving relevant information, the type and features of potential users must be considered. Designers and entrepreneurs are assumed as main targets. On this basis, a series of questions related to the architecture of the site were sought to be answered, such as:

- What will the user look for?
- How will the user search for it?
- What language would he/she most likely use?
- How should search results be illustrated and organized to best serve the user?

Accordingly, the overall structure of the portal has been defined, including how the content must be shown, searched for, and all possible actions that the user can perform once the Atlas is accessed. This stage is based on the extensive analysis of architectural web portals made in the preceding phase.

## 3. Results

### 3.1. Collection of Tactics

Up to date, 45 case studies of agri-food excellent facilities have been selected and analyzed: 28 are food factories and 17 are from the beverage section. Figure 2 shows their time distribution.

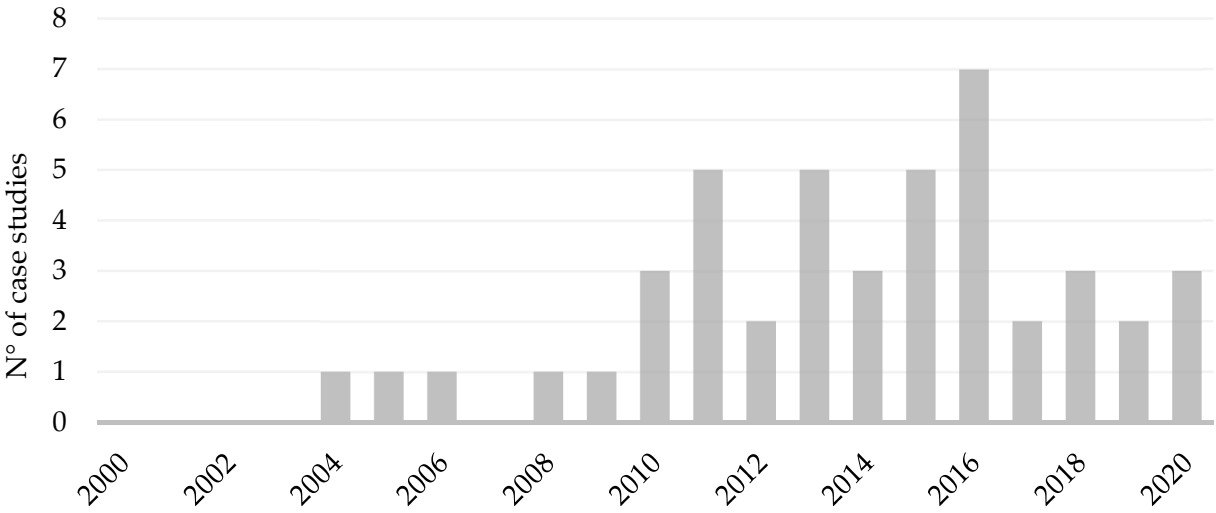

**Figure 2.** Distribution of case studies per year of construction.

From this, 106 tactics have been generated: 24 from the literature only (3 of which are agri-food specific); 29 from the case studies only, and the remaining 53 retrieved from both. The corresponding specific good practices are 295. Table 2 provides a sample list of tactics per each category. The complete list can be retrieved in [56]. Figure 3 illustrates the distribution of practices and tactics per LEED credit categories (including the perceptual-aesthetic addition).

**Table 2.** Sample list of tactics per each category.

| Category | Tactics |
|---|---|
| Integrative Process | IP 1.1 Stakeholders engagement<br>IP 1.2 Multidisciplinary design team |
| Location and Transport | LT 4.2 Urban location<br>LT 6.1 Bike racks and showers |
| Sustainable Site | SS 1.1 Site ventilation and cooling<br>SS 5.2 Green folded surfaces |
| Water Efficiency | WE 1.1 Outdoor reuse of rainwater<br>WE 2.1 Water-saving technologies |
| Energy and Atmosphere | EA 2.1 Ventilated façade<br>EA 2.5 Ground thermal proprieties |
| Materials and Resources | MR 1.2 Modular structure and/or components<br>MR 5.1 Recovery of construction and<br>demolition waste |
| Indoor Environmental Quality | EQ 2.1 Eco-friendly finishing<br>EQ 6.1 Smart lighting |
| Innovation | IN 1.2 Recovery of heat/energy from processes<br>IN 1.5 Treatment and reuse of process water |
| Regional Priorities | RP 1.1 Sharing of facilities in consortium<br>RP 1.2 Sharing of services and mixed use |
| Perceptual aesthetic aspects | PA 1.3 Landform<br>PA 4.1 Multifunction facility |

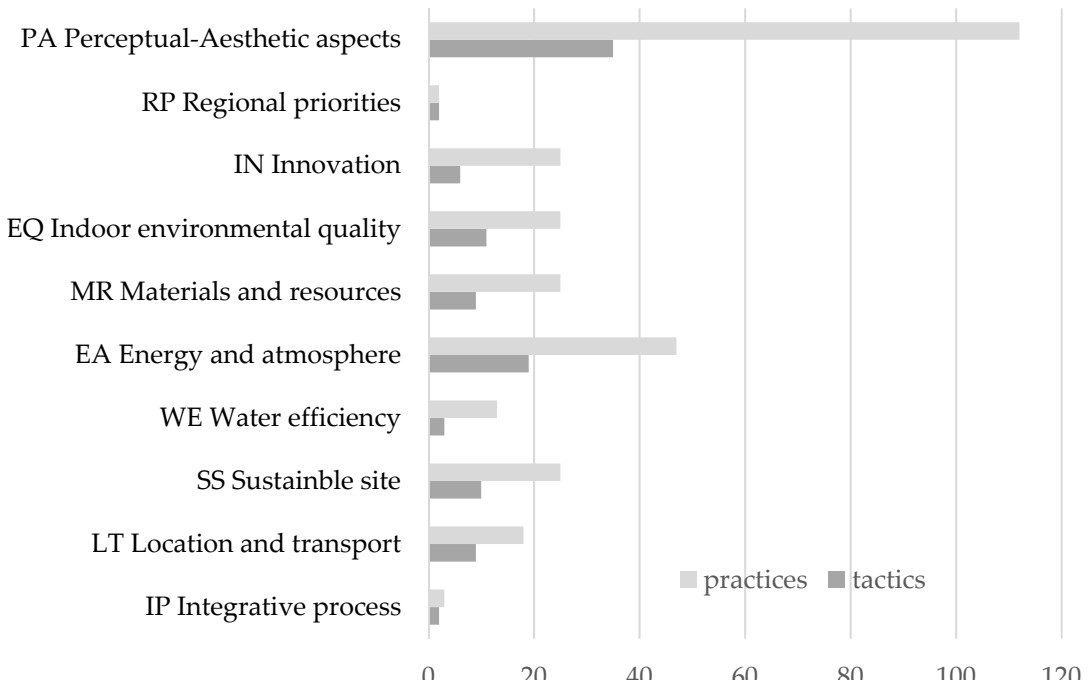

**Figure 3.** Tactics and practices distributed per topic.

## 3.2. Weighting Matrix

The weighting matrix illustrated in Section 2.2 has been applied to tactics retrieved in the first stage. Table 3 reports an extract of the implementation, where the algebraic sum of points achieved in Relative Efficacy, Benefit Duration, and Controllability of Effect per each category is shown.

**Table 3.** Weighting of tactics per Impact Categories.

| Tactic | Climate Change | Health and Well-Being | Water Resource | Biodiversity | Material Resource | Green Economy | Community |
|---|---|---|---|---|---|---|---|
| IP 1.1 Stakeholder engagement | 0 | 6 | 0 | 0 | 0 | 0 | 4 |
| IP 1.2 Multidisciplinary design team | 10 | 1 | 3 | 3 | 3 | 2 | 0 |
| LT 3.1 Building extension | 6 | 5 | 1 | 5 | 7 | 6 | 5 |

Some tactics are more "holistically sustainable", and thus reach high scores in almost all the impact categories, while others excel only in a specific category. However, there are no best tactics; rather, there are tactics that are better suited to a project based on its characteristics and the value of the firm (i.e., also the impact categories to which importance is assigned).

*3.3. Faceted Classification*

The seven facets that have been selected for the classification of tactics:

(i)    Type (i.e., Technological, Formal, functional, or management)
(ii)   Building part (e.g., Load-Bearing structure)
(iii)  Impact category (e.g., Climate change)
(iv)   Material (e.g., Wood)
(v)    Type of intervention (i.e., new construction, renovation, or suitable to both)
(vi)   General strategy (e.g., energy efficiency)
(vii)  Level (i.e., site, building, process, or not scale-dependent)

Table 4 shows part of them and their attributes. Not Defined (N.D.) or Not Applicable (N.A.) were used when any specific attributes were selectable and when the category was irrelevant to the tactic, respectively.

**Table 4.** Extract of facets and attributes of the classification system.

| (ii) Building Part | (iii) Impact Category | (iv) Material | (vi) General Strategy |
|---|---|---|---|
| Load-bearing structure | Climate change | Wood | Energy efficiency |
| Vertical envelope | Health and well-being | Concrete | Renewable resources |
| Top horizontal envelope | Water resource | Metals | Bioclimatic |
| Low horizontal envelope | Biodiversity | Glass | Sustainable mobility |
| Internal partition | Material resource | Coating and paint | Community engagement |
| Systems | Green economy | Vegetation a/o water | Well-Being of workers |
| Outdoor equipment | Community | | Visual mitigation |

Each tactic was then classified through these categories and attributes. The classification was reported in the spreadsheet where tactics are described (phase 1). This allows data to be easily retrieved to both feeding a digital atlas in the web and physical cards of tactics.

*3.4. Architecture of the Digital Atlas*

The last phase of the study led to defining the architecture of the information for the digital atlas, trying to answer the questions asked in Section 2.4.

- Firstly, the content to be shown once the user accesses the atlas has been defined: tactics are the primary object of the search, then associated practices and case studies can be illustrated and references to bibliography or webography provided.
- The user will probably use keywords or access interesting categories to search tactics. Hence, the search interface might combine an SQI—as in many architectural portals—with a horizontal menu including multiple choices or filters. The two modes will allow users to follow the most suitable for them.
- The SQI will make use of an information-retrieval system linking typed words with synonyms or semantic fields connected to available tactics. An automatic filling tool for words can also be implemented to help non technicians.

- Results of the user search must be clear to both designers and entrepreneurs. Most relevant data should be highlighted, and the shift to linked information (tactics or supplementary materials) must be easy. Possibly, the user should be able to track the browsing and save preferred contents (either on the digital portal or by downloading a pdf version). Correlated tactics must be highlighted to the user basing the selection on the type of search (attributes or categories) the user is interested in.

Figure 4 drafts a conceptual visualization of two pages: the Home Page and a sample of the Tactic Page.

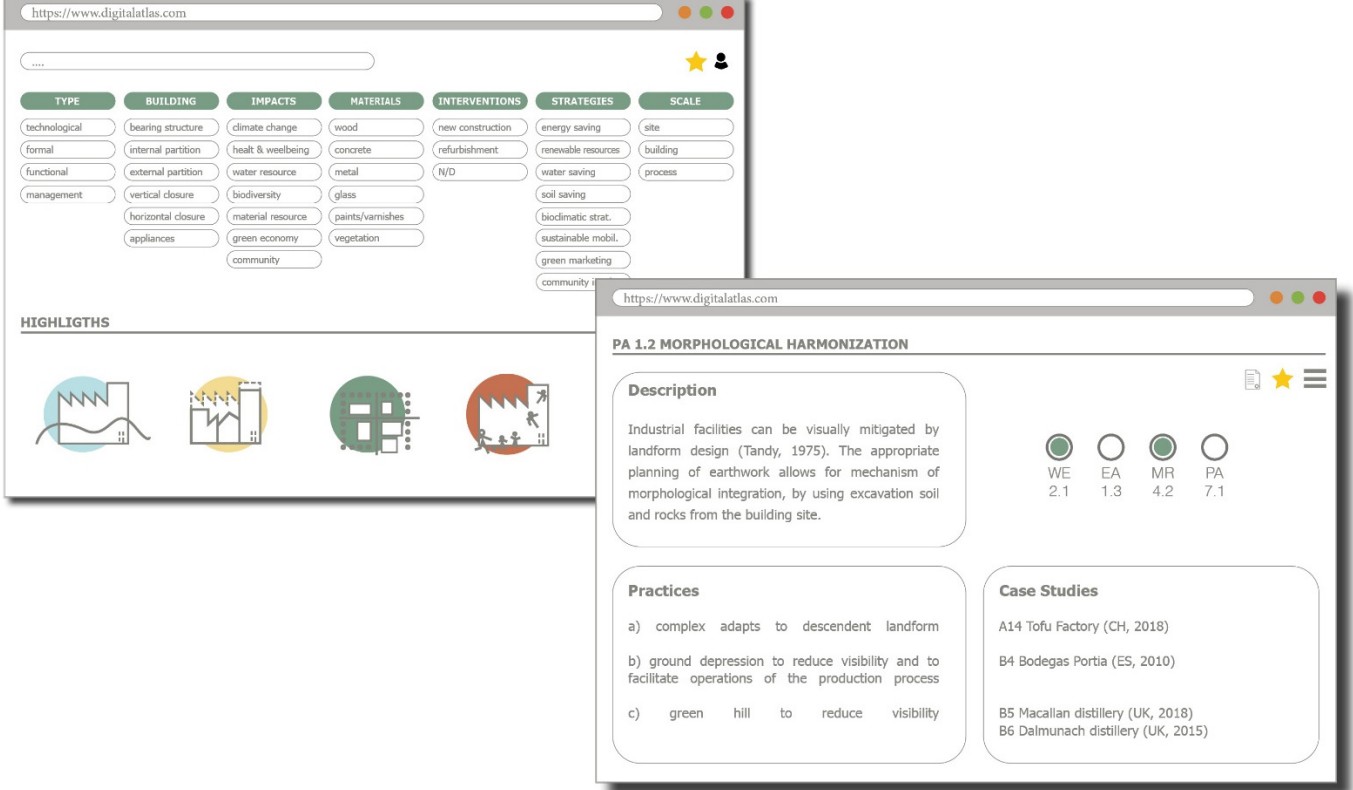

**Figure 4.** Conceptual visualization of two sample pages of the Atlas.

### 3.5. Possible Implementation of the Atlas

According to the premises, the Atlas implementation is devolved to each potential user, who can operate freely because no predefined actions are set when she/he enters. The navigation is driven by the user's goal as well as the established project requirements. However, the most likely implementation modality the authors expect to be adopted is that the user selects one or more attributes referring to one or more facets among those available on the Home Page, based on the type of mitigation she/he seeks.

This process can be facilitated if the user has previously implemented the assessment system developed by the authors and outlined in Section 1.2, as this will have highlighted the main project criticalities that must be mitigated. However, the Atlas can be used regardless of the assessment system.

Aiming at providing the reader with a practical example of this implementation, the authors tested the Atlas in a case study that had already been analyzed by using the System during the first phase of the research project.

The case study examined a recently constructed industrial facility in Cesena (Italy) owned by a leading agri-food firm in the country. The project's evaluation stage revealed that several good environmental practices were already implemented in contrast to an extremely low scenic integration of the site and social attractiveness. So, the authors have simulated the attitude of a potential user of the Atlas in this situation. Thus, they have

primarily looked for strategies at the site level that are helpful in enhancing the architectural image of the building while providing a pleasant outdoor space for workers and possibly producing further environmental positive effects. Figure 5 illustrates the obtained mitigation scenario, with the codes in the colored balls representing the chosen tactics.

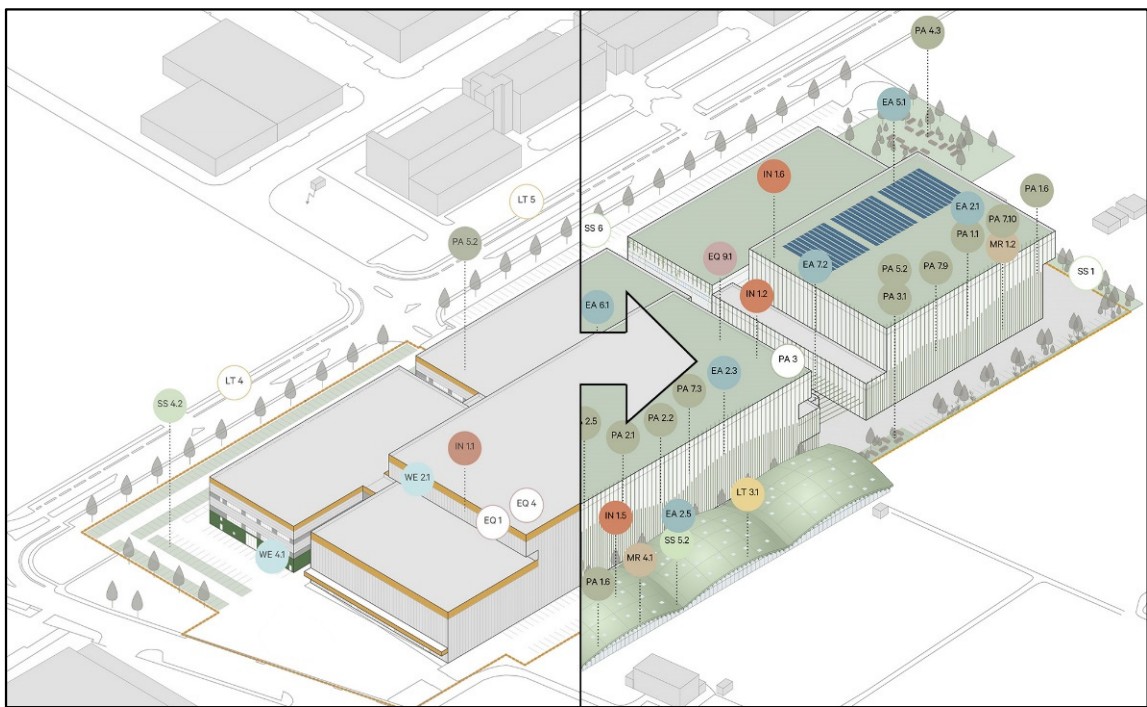

**Figure 5.** Scenarios before (**left**) and after (**right**) the implementation of the Atlas.

## 4. Discussion

The number of case studies retrieved in the first stage demonstrate that the need for holistic, sustainable design of factories is not just something asked for by policymakers and scientists, but a fact that has been gradually understood by firms themselves. The progression in the number of exemplary projects during the last twenty years (time distribution in Figure 2) reinforces this thesis. Thus, the Atlas intends to assist users in shifting from current knowledge segmentation toward a more integrated and holistic view of sustainability. In this sense, the overall project seeks to fill the gap in building assessment and design-support tools, such as the GBRSs that are environmentally focused [26].

Moreover, according to OECD, Kaur et al., and Despeisse [9,19,57], one of the main barriers limiting enterprises from designing sustainable manufacturing is a lack of knowledge and easy-to-understand information regarding appropriate mitigation actions and their related benefits. Even the most widely used GBRSs are more focused on the evaluation stage than the design-support phase. To that goal, the study has offered a set of design strategies that are easy to understand because they are briefly, but thoroughly, discussed and linked to real-world case studies from which they are inspired. The list of case studies, practices, and even tactics can certainly expand in the future, but it can already be useful material to getting started. In fact, the over a hundred tactics address a wide range of issues faced by designers and businesses nowadays, from location and transportation to the factory's perceptual-aesthetic impact on the landscape.

Because even when design tactics or tools are available, companies fail to grasp their impact [58] and the potential benefits of their actions on the landscape, a weighted matrix for tactics has been defined. This may assist users in finding the most appropriate collection of tactics in the Atlas based on a preferred impact category, or in selecting tactics that have a positive impact in more than one category and hence are more "holistically sustainable".

According to the premises, these latter should deliver the most value to both the firm and the territory, leading towards the "positive impact factory" [2].

A faceted classification has been developed with the goal of supporting the adoption of excellent practices from several landscape dimensions. By following the categorization, users are aided in selecting appropriate approaches. For example, if users choose bioclimatic strategies as an attribute of the general strategies category, the Atlas will return all relevant tactics. Then, based on what the user is looking for, whether it is a complementary or additional strategy, other categories can be selected, such as all those working on the top horizontal closure, and so forth. This is expected to be highly valuable to assist teams in applying System Thinking and Integrated Design approaches as recommended by the U.S. GBC [37].

Lastly, since many available design-support tools are seen as overcomplicated and not user-friendly, the Atlas has been conceived as a digital, accessible, and easy-to-navigate tool. As a matter of fact, it represents a novelty in the field of architecture portals, whose search is usually case-driven as discovered in the investigation presented in Section 2.3. Instead, the Atlas is tactic driven, so the user is offered distilled and elaborated information—that is a general tactic—from which in turn he/she might backtrack to the particular case study.

Furthermore, because everything is digital, integrations are always feasible. Given that the methodology is iterative, every change or addition would necessitate updating the Atlas. Depending on the nature of the news, this could imply introducing a single tactic card or updating all cards due to the addition of a new facet. In any case, the general portal can continue to function.

Concerning limitations, despite concentrating on the agri-food sector, the results of the research are simple to adapt to other sectors. The credits for the assessment system are general enough to be applied in a variety of different industrial buildings, and ad hoc requirements are given when needed for agri-food. Furthermore, the tactics are also a generalization of specific practices, and in most cases can be applied regardless of the function of the site.

This paper presented the Atlas in its genesis, but it still must be developed as "real" web portal. For this reason, a validation stage of the Atlas could be set for future developments of the study through, for example, a survey on a large sample of potential users (i.e., designers and businesses). It would also be interesting to open the discussion with participants to receive their suggestions for new tactics or facets to be added. This would provide an idea of how it could be if the Atlas was coproduced with stakeholders.

## 5. Conclusions

The article illustrated that a holistic, sustainable approach to factory design is needed, because both policymakers and academics argue that this can benefit both the territory and the firm. However, available assessment and design-support tools adopt sectorial approaches that fail in achieving this goal. Therefore, these are often ineffective in informing the user (i.e., businesses and designers) about positive effects due to a combination of environmental strategies with social and perceptual-aesthetic ones.

This paper hence outlined a research project that started in 2016 with the aim of filling some gaps in tools for designing more sustainable factories. The main novelty of the study is to address the multi-faceted impact of factories on the landscape through one tool, thus attempting to consider all relevant aspects of the context and maximize positive synergies among them. In particular, the article focused on a part of this study, which attempts to provide designers with a rich catalogue of design tactics to mitigate the impact of factories on the landscape. Hence, the assumptions and methodology to develop a Digital Atlas of Tactics has been presented: from the collection of case studies from which, in turn, retrieving design tactics, to the definition of a set of impact categories for weighting their effectiveness; from the classification of tactics via a multi-faceted labelling system to the definition of the architecture of the web site (i.e., the Digital Atlas). The digital format of

this outcome represents another original element of the study, which allows a dynamic, easy, and smart implementation of the toolkit for designing sustainable factories.

Although new case studies, practices, tactics, and even facets of the classification system can be added at any point, the results that have been presented so far might already be useful to designers and businesses that want to reduce the impact of their facilities on the landscape. The structure of the tool, especially thanks to the faceted classification, makes connections and synergies among tactics easy. Therefore, the Atlas goes right in the direction desired by the European Commission through the NEB: sustainable, beautiful, and together. The Atlas is structured as complex but not complicated, as one of the premises was that firms need to be informed about possible mitigation actions and due benefits in an easy and clear manner. For this reason, the architecture of the information has been conceived to be flexible, and capable to adapt to needs and preferences of different users.

**Author Contributions:** Conceptualization, L.M. and E.A.; methodology, L.M. and E.A.; investigation, L.M.; resources, L.M.; writing—original draft preparation, L.M.; writing—review and editing, E.A.; visualization, L.M.; supervision, E.A. All authors have read and agreed to the published version of the manuscript.

**Funding:** This research received no external funding.

**Institutional Review Board Statement:** Not applicable.

**Informed Consent Statement:** Not applicable.

**Data Availability Statement:** Not applicable.

**Conflicts of Interest:** The authors declare no conflict of interest.

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
