# Peer review of "Digital Atlas of Tactics to Designing Sustainable Factories"

_sustainability, doi:10.3390/su14074321_

Round 1
Reviewer 1 Report
This paper provides a holistic sustainable approach to factory design through a digital Atlas of tactics to mitigate the negative impacts of factory design on the landscape. The research topic has practical significance. However, there are still many areas that need to be further improved. The specific opinions are as follows.
- In the introduction, the introduction of the research background can be reduced, the content of the current research status can be increased, and a general review of the existing research should be made to highlight the necessity and advantages of this research.
- Line 132 on Page 3. Was this research initiated by the authors or by other researchers? If the former, please add more information about how this research was conducted. In the latter case, a corresponding citation or reference should be added.
- The analysis of the results can be further in-depth. Is it possible to visually demonstrate the impact of the plant design on the landscape using the digital Atlas of tactics?
- The innovation of the paper is not highlighted, please explain.
Author Response
This paper provides a holistic sustainable approach to factory design through a digital Atlas of tactics to mitigate the negative impacts of factory design on the landscape. The research topic has practical significance. However, there are still many areas that need to be further improved. The specific opinions are as follows.
# RESPONSE The authors are grateful for the time spent in this revision and for pointing out the required changing to improve the overall quality of the paper. The authors did they best to properly embed the reviewer’s suggestions in the manuscript.
- In the introduction, the introduction of the research background can be reduced, the content of the current research status can be increased, and a general review of the existing research should be made to highlight the necessity and advantages of this research.
# RESPONSE The background of the research has been reduced (Line 24 to 82).
The content and status of the current research has been increased by adding some specifications and clarifications (e.g., New figure 1) and an entire new paragraph showing a possible output of the Atlas implementation (3.5).
For what concern the general review of existing research, as it is stated in the article it comes of an emerging topic which has limited specific literature, but a promising future as reinforced by the New European Bauhaus vision. The few existing studies are mentioned in the introduction and practical approaches in the paragraph of existing tools (par. 1.1.), where their flaws are discussed and the necessary advanced introduced.
- Line 132 on Page 3. Was this research initiated by the authors or by other researchers? If the former, please add more information about how this research was conducted. In the latter case, a corresponding citation or reference should be added.
# RESPONSE The study has been entirely performed by the authors in two major stages: the first has been already explained in previous publications, which are mentioned at lines 165 and 172. However, for more clarity, this have been made explicit at lines 137-139 and the diagram of these research phases has been included in the paper (lines 143-145 and new Figure 1).
- The analysis of the results can be further in-depth. Is it possible to visually demonstrate the impact of the plant design on the landscape using the digital Atlas of tactics?
# RESPONSE It has been added a paragraph (3.5) in the results section in order to outline how the Atlas can help the architect to select useful and synergic tactics and thus aiming at designing a holistically sustainable factory.
- The innovation of the paper is not highlighted, please explain
# RESPONSE A sentence has been added to the abstract to make it evident from the beginning what the study's novelty is (line 14). At lines 140-143 and 498-501 it is also illustrated the main novelty of the research project, and at lines 470-472; 507-509 it has been highlighted the originality of the proposed tool, namely the Atlas, compared to less comprehensive, dynamic and easy to apply design support tools for factories.

Reviewer 2 Report
Dear authors,
Thank you for writing the article that particularly addresses the issue of factory design. In the abstract, you mention about other possible approaches in factory designs other than a digital atlas toolkit. It would be great f you could mention some of them.
Please consider adding a diagram of the research phases and stages in the beginning. Each of the phases may be added with metrics of some sort of evaluation.
Also, in Section 2.2, you mention "... to support the design and guide users in mitigating the impact of their factories." It will be helpful if you could provide examples of how Atlas is being implemented.
In the result section, can you describe in detail 1-2 case studies using the Atlas?
Author Response
Thank you for writing the article that particularly addresses the issue of factory design.
# RESPONSE The authors are grateful for the time spent in this revision and for pointing out the required changing to improve the overall quality of the paper. The authors did they best to properly embed the reviewer’s suggestions in the manuscript.
In the abstract, you mention about other possible approaches in factory designs other than a digital atlas toolkit. It would be great if you could mention some of them.
# RESPONSE The other available tools and their flaws are presented in paragraph 1.1, and this is also anticipated at line 76. However, we added very few words at Lines 76 and 84 to make the intent more evident since the first sight. It comes of environmental assessment tools like GBRSs and visual impact guidelines. In addition, also conventional handbooks are mentioned. The argument is that despite being helpful, all these tools and approaches fail in addressing sustainable factory design in a holistic way.
Please consider adding a diagram of the research phases and stages in the beginning. Each of the phases may be added with metrics of some sort of evaluation.
# RESPONSE The two main phases of the whole research project have been visually synthetized in the new Figure 1, where the reader can also find the specific aim, method, and outcome of each performed activity.
Also, in Section 2.2, you mention "... to support the design and guide users in mitigating the impact of their factories." It will be helpful if you could provide examples of how Atlas is being implemented.
In the result section, can you describe in detail 1-2 case studies using the Atlas?
# RESPONSE It has been added a paragraph (3.5) in the results section to outline how the Atlas can help the architect to select useful and synergic tactics and thus aiming at designing a holistically sustainable factory. In addition, a mitigation scenario for a case study is shown in comparison with the original project.

Round 2
Reviewer 1 Report
The paper is well revised. However, the following items can be further improved.
(1) The introduced should be added the latest references.
(2) The novelty of the paper should be listed more clear.
(3) More comparisons are needed in the Discussion section.
Author Response
The authors did they best to properly embed the reviewer’s further suggestions in the manuscript.
In particular:
- The introduced should be added the latest references.
# RESPONSE
Further references on the topic have been added to section 1. They are an article by ARUP on green factories (2022); a study by Bartolucci (2022) on the concept of Net Zero Energy Factory; a study by Wang (2019) on sustainable industrial site redevelopment planning support system; a toolkit by Gothmann at al. (2015) on Sustainable Industrial Areas and an investigation by the center of Excellence on Green Productivity (2013).
- The novelty of the paper should be listed more clear.
# RESPONSE
Authors have tried to highlight more clearly the novelty of the paper. To this end:
- an additional sentence about the originality of the overall study was added at lines 124-125.
- the original key features of the Atlas are now listed at the end of section 1 (lines 178-183)
Beyond these points, the novelty of the study is already mentioned in the abstract; discussion and conclusion sections as illustrated for the previous revision.
- More comparisons are needed in the Discussion section.
# RESPONSE
In the discussion section now the reader can find direct reference to previous studies and the position of this research in comparison. These have been added at lines 430-434; 437-438; 446; 451; 458-460.